# LncRNA NR_003508 Suppresses *Mycobacterium tuberculosis*-Induced Programmed Necrosis via Sponging miR-346-3p to Regulate RIPK1

**DOI:** 10.3390/ijms24098016

**Published:** 2023-04-28

**Authors:** Li Liu, Zhirui Yu, Qinmei Ma, Jialin Yu, Zhaoqian Gong, Guangcun Deng, Xiaoling Wu

**Affiliations:** 1Key Lab of Ministry of Education for Protection and Utilization of Special Biological Resources in Western China, Ningxia University, Yinchuan 750021, China; l13895378870@163.com (L.L.); 18309671975@163.com (Z.Y.); aries6687@163.com (J.Y.); gong_zhaoqian@163.com (Z.G.); 2School of Life Science, Ningxia University, Yinchuan 750021, China

**Keywords:** LncRNA NR_003508, programmed necrosis, *Mycobacterium tuberculosis*, miR-346-3p

## Abstract

Emerging evidence suggests that long non-coding RNAs (LncRNAs) are involved in Mtb-induced programmed necrosis. Among these LncRNAs, LncRNA NR_003508 is associated with LPS-induced acute respiratory distress syndrome. However, whether LncRNA NR_003508 contributes to Mtb-induced programmed necrosis remains undocumented. Firstly, the expression of LncRNA NR_003508 was determined using RT-qPCR and FISH. The protein expression of RIPK1, p-RIPK1, RIPK3, p-RIPK3, MLKL, and p-MLKL was measured by Western blot in RAW264.7 and mouse lung tissues. Furthermore, luciferase reporter assays and bioinformatics were used to predict specific miRNA (miR-346-3p) and mRNA (RIPK1) regulated by LncRNA NR_003508. In addition, RT-qPCR was used to detect the RIPK1 expression in TB patients and healthy peripheral blood. The flow cytometry assay was performed to detect cell necrosis rates. Here we show that BCG infection-induced cell necrosis and increased LncRNA NR_003508 expression. si-NR_003508 inhibited BCG/H37Rv-induced programmed necrosis in vitro or in vivo. Functionally, LncRNA NR_003508 has been verified as a ceRNA for absorbing miR-346-3p, which targets RIPK1. Moreover, RIPK1 expression was elevated in the peripheral blood of TB patients compared with healthy people. Knockdown of LncRNA NR_003508 or miR-346-3p overexpression suppresses cell necrosis rate and ROS accumulation in RAW264.7 cells. In conclusion, LncRNA NR_003508 functions as a positive regulator of Mtb-induced programmed necrosis via sponging miR-346-3p to regulate RIPK1. Our findings may provide a promising therapeutic target for tuberculosis.

## 1. Introduction

Over the past two decades, although people have made concerted efforts to develop new diagnostic methods, drugs, and vaccines and continuously expanded research and development channels, tuberculosis is still one of the major infectious diseases causing death in the world [1]. An estimated 10 million cases remained undiagnosed and uncured, resulting in 1.3 million deaths annually [2]. TB is principally related to the ability of macrophages to eliminate pathogenic bacteria. Previous studies have shown that the host cell death mode is the main factor affecting this process [3,4]. In particular, the spread of bacteria was enhanced by necrotic death, which was regulated by many factors, including Long non-coding RNA (LncRNA) [5]. For example, a new LncRNA, LncCRLA, inhibited RIPK1-induced necrosis by blocking the RIPK1-RIPK3 interaction [6].

Being endogenous RNA over 200 nucleotides long and without a potential open reading frame, LncRNA plays a pivotal role in the regulation of gene expression at multiple levels [7]. Additionally, it additionally regulated molecules by multiple mechanisms, including cell cycle, autophagy, apoptosis, and programmed necrosis [8,9,10,11]. Functionally, LncRNAs could regulate the expression of downstream target genes through miRNAs. One striking example was that LncRNA NRF regulated programmed necrosis and myocardial injury during ischemia and reperfusion by targeting miR-873 [12].

Programmed necrosis was considered a type of regulated cell death mediated by receptor-interacting serine/threonine-protein kinase 1 (RIPK1), receptor-interacting serine/threonine-protein kinase 3 (RIPK3), and mixed lineage kinase domain-like (MLKL). According to recent studies, inhibition of focal adhesion kinase (FAK) increased necrosis occurring in a RIPK1-dependent manner throughout Mtb infection [13]. Additionally, B-cell lymphoma-extra large (Bcl-XL) mediated RIPK3-dependent necrosis in Mtb-infected macrophages [14]. These studies confirmed the importance of necrosis-key genes in the process of programmed necrosis. Evidence has been conferred that the necrosis key genes are regulated by LncRNAs. LncRNA H19 served as a sponge of miR-103/107 regulation of the expression of FADD, which attenuated necrotic responses by influencing the formation of RIPK1 and RIPK3 complexes in H9c2 cells [15]. Our previous study has found that LncRNA NR_003508 is involved in LPS-induced acute respiratory distress syndrome [16]. However, the function of LncRNA NR_003508 in Mtb-induced macrophage programmed necrosis is still unclear. Thus, it is vital to explore and reveal the LncRNA NR_003508 molecular mechanism underlying the regulation of programmed necrosis.

This study determined the function of LncRNA NR_003508 in the process of Mtb-induced programmed necrosis. Mechanistically, LncRNA NR_003508 served as an endogenous sponge for miR-346-3p to regulate RIPK1 expression. Given their critical role in regulating TB, LncRNA NR_003508-miR-346-3p-RIPK1 axes are supposed to be a potential therapeutic target in the future.

## 2. Results

### 2.1. BCG-Infection Stimulates Cell-Programmed Necrosis Accompanies with Increased LncRNA NR_003508 Expression

Being a vaccine strain of tuberculosis, BCG could induce cell-programmed death, such as apoptosis and necrosis. Qin Sun et al. confirmed that Mtb infection led to programmed necrosis in human macrophages, inflicting CypD-p53-ANT1 mitochondrial association and mitochondrial depolarization [17]. In order to explore the mechanism of cell necrosis, the necrosis rate of BCG-infected RAW264.7 cells was measured by immunofluorescence assay. As shown in Figure 1A, the necrosis rate was significantly increased with 10 CFU of BCG infections. Therefore, we chose 10 CFU for subsequent experiments. We next evaluated the necrotic ultrastructural changes of necrosis in RAW264.7 cells infected with BCG at 18 h. As shown in Figure 1B, RAW264.7 cells exposed in BCG for 18 h presented a typical necrotic cell death morphology with extensive vesiculation of cytoplasmic organelles, rupture of the cell membrane integrity, and release of cellular contents. As the markers of programmed necrosis, p-RIPK1/RIPK1, p-RIPK3/RIPK3, and p-MLKL/MLKL expression were measured by Western blot. The result showed that the expression of p-RIPK1/RIPK1, p-RIPK3/RIPK3, and p-MLKL/MLKL in RAW264.7 cells were dramatically up-regulated by BCG infection and peaked at 18 h (Figure 1C).

Subsequently, we used qRT-PCR to detect the cellular LncRNA NR_003508 mRNA expression. As displayed in Figure 1D,E, compared to the control, varying the doses of BCG infection remarkably up-regulated LncRNA NR_003508 expression in RAW264.7 cells, which were also increased by BCG infection in a time-dependent manner. Furthermore, fluorescence in situ hybridization (FISH) confirmed that compared with the control, the expression of LncRNA NR_003508 was accumulated by BCG infection at 18 h (Figure 1F). Combined with the previous results, we optimized the time and dose of BCG with 10 CFU and 18 h for subsequent experiments. Since RNA expression was up-regulated after BCG infection, we constructed three small interference RNAs (siRNA-1, siRNA-2, and siRNA-3) to silence LncRNA NR_003508 in RAW264.7 cells. As shown in Figure 1G, siRNA-2 significantly reduced the LncRNA NR_003508 expression compared with the NC group. We then used qRT-PCR to confirm the interference of LncRNA NR_003508 in BCG-treated RAW264.7 cells (Figure 1H). The above data confirmed that we successfully constructed the interference model.

### 2.2. si-NR_003508 Attenuates Mycobacterium Tuberculosis Induced Programmed Necrosis In Vitro and In Vivo

To verify whether LncRNA NR_003508 regulated the programmed necrosis, TEM was used to observe the morphological structure changes of RAW264.7 cells. It could be found from Figure 2A that BCG-infected cells showed typical necrotic characteristics with broken cell membranes and shrinkage of the nucleus, while si-NR_003508 mitigated this phenomenon. Furthermore, as shown in Figure 2B, compared with the NC group, the cell necrosis rate was significantly up-regulated in the BCG-infected group. Conversely, si-NR_003508 reversed this phenomenon. Of note, the accumulation of reactive oxygen species (ROS) induced cell necrosis. Therefore, we used CellROX Orange Reagent to detect the intracellular ROS. As Figure 2C displays, BCG-infected cells remarkably accumulated ROS compared to the NC group, and si-NR_003508 reduced BCG-induced ROS accumulation in RAW264.7. Then, the C57BL/6J mice were intratracheally instilled with BCG at 3 × 10^6^ CFU/mouse at a specific time, as Figure 2D displayed, and we found that BCG infection decreased the body weight of mice in the observation period, which was exacerbated by si-NR_003508 (Figure 2E). This implied that si-NR_003508 might attenuate the lung injury caused by BCG infection in mice. We further monitored the effect of si-NR_003508 on lung injury in mice by intratracheal instillation. The result showed that the BCG-infected mouse lung exhibited vascular congestion and bleeding. Moreover, there was increased infiltration of inflammatory cells in the BCG group; intriguingly, si-NR_003508 significantly alleviated the symptoms (Figure 2F). The results indicated that si-NR_003508 could alleviate BCG-induced cell death and lung injury.

To further confirm the effect of LncRNA NR_003508 on BCG-induced programmed necrosis, we used qRT-PCR to validate the expression of RIPK1, RIPK3, and MLKL in BCG-infected RAW264.7 cells. As can be noted in Figure 3A, compared with the NC group, si-NR_003508 effectively reduced the expression of RIPK1, RIPK3, and MLKL, as well as p-RIPK1/RIPK1, p-RIPK3/RIPK3, and p-MLKL/MLKL in RAW264.7 cells (Figure 3B). A similar result was collected from mouse lung tissues (Figure 3C). Finally, we infected RAW264.7 cells with H37Rv. The results showed that the expression of p-RIPK1/RIPK1, p-RIPK3/RIPK3, and p-MLKL/MLKL was increased with H37Rv infection, while si-NR_003508 effectively reduced the expression of p-RIPK1/RIPK1, p-RIPK3/RIPK3, and p-MLKL/MLKL (Figure 3D). With this in mind, we drew the conclusion that LncRNA NR_003508 played an important role in regulating programmed necrosis.

### 2.3. LncRNA NR_003508 Functions as a Competing Endogenous RNA by Binding to miR-346-3p 

Emerging lines of evidence have reported that cytoplasm LncRNAs could function as competing endogenous RNAs (ceRNAs) by binding to and sequestering specific miRNAs that blocked target gene expression [18,19]. Given that LncRNA NR_003508 is mainly located in the cytoplasm, we hypothesized that LncRNA NR_003508 could serve as sponges for microRNAs in BCG-infected RAW264.7. Consequently, we screened six miRNAs such as miR-346-3p, miR-34a-5p, miR-320-5p, miR-212-5p, miR-152-5p, and miR-301b-3p, with high scores related to programmed necrosis and LncRNA NR_003508 based on multiple bioinformatics databases (Figure 4A). As shown in Figure 4B, compared with the control, among the six miRNAs, the expression of miR-346-3p in cells was decreased in 10 CFU of BCG. However, si-NR_003508 significantly increased the expression of miR-346-3p. Hence, we synthesized the miR-346-3p mimic for subsequent research. According to the results displayed in Figure 4D, the expression of miR-346-3p was highly up-regulated with transfection of the miR-346-3p mimic, which indicated that we had successfully established the miR-346-3p overexpression model. For further investigation, we used the Bielefeld Bioinformatics Service (https://bibiserv.cebitec.uni-bielefeld.de/rnahybrid/), accessed on 10 July 2019, to predict the binding site between LncRNA NR_003508 and miR-346-3p (Figure 4E). Then, the plasmid was constructed based on the binding sites of the luciferase reporter assay. Additionally, we found that the luciferase activity of LncRNA NR_003508-WT was inhibited by miR-346-3p, indicating LncRNA NR_003508 functioned as a ceRNA sponging miR-346-3p and regulating miR-346-3p expression (Figure 4F). Concomitantly, immunofluorescence assays confirmed the same results in RAW264.7 cells. (Figure 4G). In this regard, it was suggested that LncRNA NR_003508 functioned as a ceRNA and regulated miR-346-3p expression.

### 2.4. miR-346-3p Targets RIPK1 to Regulate Programmed Necrosis

Given LncRNA NR_003508 regulated necrosis by binding to miR-346-3p, we further screened the potential mRNAs related to programmed necrosis and interacted with miR-346-3p by using the online prediction software TargetScan 8.0 (https://www.targetscan.org/vert_80/), which accessed on 29 September 2021 (Figure 5A). As presented in Figure 5B, the luciferase activity of RIPK1-WT was reduced by the miR-346-3p mimic, while RIPK1-MUT had no obvious change, indicating a direct interaction between RIPK1 and miR-346-3p. Moreover, data from Figure 5C,D illustrated that both BCG and H37Rv enhanced RIPK1 expression, which was suppressed by miR-346-3p mimic. Additionally, an accordant result in immunofluorescence was observed in Figure 5E. Considering the negative correlation between LncRNA NR_003508 and miR-346-3p in BCG-infected RAW264.7 cells, we further wondered whether LncRNA NR_003508 regulates the expression of the miR-346-3p target gene RIPK1. Thus, immunofluorescence was used to measure the RIPK1 expression of RAW264.7 cells at the morphological level. Additionally, compared with the BCG group, the si-NR_003508+BCG infected group significantly decreased the expression of RIPK1 (Figure 5F). Moreover, si-NR_003508 prominently restrained RIPK1 protein expression with BCG and H37Rv infection was observed in Figure 5G,H. In addition, a striking result showed that RIPK1 expression in the peripheral blood of TB patients was significantly elevated compared with healthy people (Figure 5I). These results suggested that LncRNA NR_003508 augmented RIPK1 expression in RAW264.7 through the miR-346-3p/RIPK1 signaling axis.

### 2.5. miR-346-3p Participates in Programmed Necrosis through RIPK1/RIPK3/MLKL Axis

In order to further clarify the mechanism of miR-346-3p in BCG-induced necrosis, we delved into the necrosis rate, ROS accumulation, and Ca^2+^ influx of RAW264.7 cells infected by BCG. Just as Figure 6A,B depicted BCG elevated cell necrosis rates and ROS accumulation, which were mitigated by miR-346-3p mimic. Of note, overexpression of miR-346-3p could significantly reduce the Ca^2+^ concentration in BCG-infected macrophages, indicating that miR-346-3p mimic could inhibit the accumulation of Ca^2+^ concentration in BCG-infected macrophages, thereby inhibiting programmed necrosis (Figure 6C). Notably, RIPK1 and RIPK3 kinases played a central role in TNF-induced programmed necrosis [20]. Our data implied LncRNA NR_003508 was involved in BCG-infected cell necrosis via sponging miR-346-3p; moreover, BCG induced programmed necrosis through RIPK1/RIPK3/MLKL. In conjunction with the previous literature, we predicted that miR-346-3p might embroil in cell programmed necrosis with BCG infection by regulating the RIPK1/RIPK3/MLKL axis. As evidenced in Figure 6D, BCG evidently induced RIPK1, RIPK3, and MLKL phosphorylation which was reversed by miR-346-3p mimic. We further verified the results with H37Rv infection, as shown in Figure 6E; H37Rv promoted the expression of p-RIPK1/RIPK1, p-RIPK3/RIPK3, and p-MLKL/MLKL which were reversed by miR-346-3p mimic.

Collectively, a schematic diagram of LncRNA NR_003508 regulating RIPK1-mediated programmed necrosis pathway via miR-346-3p was presented in Figure 6F. BCG and H37Rv induced the overexpression of LncRNA NR_003508, which functioned as a miR-346-3p sponge to positively regulate RIPK1 expression in RAW264.7, implying that LncRNA NR_003508 interacted with miR-346-3p modulated programmed necrosis through RIPK1/RIPK3/MLKL. Subsequently, the phosphorylated RIPK1, RIPK3, and MLKL promoted the ROS accumulation of mitochondria and Ca^2+^ influx in RAW264.7 cells, further exacerbating cell necrosis and persistent infection. In summary, these results elucidated that LncRNA NR_003508 activated the RIPK1/RIPK3/MLKL pathway by sponging miR-346-3p.

## 3. Discussion

As an intracellular pathogen, Mtb has coevolved with humans for several years, which not only adapts to the host environment but also interferes with various cellular functions of the host to inhibit its killing by the host cells [21,22]. A recent study additionally confirmed that protein kinase G (PknG) exhibited both ubiquitin-activating enzyme (E1) and ubiquitin ligase (E3) enzyme activities to block phagosome fusion and pathogen clearance, thereby weakening the innate host responses during Mtb infection [23,24]. Additionally, Mtb infection could also cause programmed necrosis, thereby establishing persistent infections [17]. Here we found that BCG could induce RAW264.7 cells programmed necrosis, which was consistent with previous studies [25]. Moreover, previous studies manifested that ceRNAs regulated networks involved in various biological activities [26,27]. This study investigated a novel model in which the LncRNA NR_003508 promoted the activation of the RIPK1/RIPK3/MLKL pathway via sponging miR-346-3p.

Increasing evidence instructed that LncRNAs have acted as key regulatory factors in various biological processes, such as cell proliferation, differentiation, autophagy, and apoptosis, which have significant implications for understanding diseases [28]. Preceding studies indicated that lncRNA Hox transcript antisense intergenic RNA (HOTAIR) aggravated autophagy and promoted apoptosis of nucleus pulposus cells. More strikingly, LncRNA small nucleolar RNA host gene 7 (SNHG7) played an important role in osteoarthritis apoptosis and autophagy via sponging miR-34a-5p [29]. Additionally, it has been confirmed that Linc00870 regulated Th1 and Th2 cells by JAK/STAT pathway in peripheral blood mononuclear cells infected by Mtb [30]. Additionally, TNF-α and hnRNPL-related immunoregulatory lncRNA (THRIL) adjusted the expression of TNF-α in human monocytes to drive transcription following TLR2 activation [31]. In the current study, we found that the expression of LncRNA NR_003508 was significantly up-regulated with BCG infection. Further research revealed that the LncRNA NR_003508 level was displayed in a time-dependent manner with BCG infection, indicating that LncRNA NR_003508 could play a central role in immune regulation.

According to previous studies, LncRNAs also regulated programmed death. For instance, ectopic overexpression of LncRNA EPIC1 (“Lnc-EPIC1”, ENSG00000224271) inhibited Dexamethasone (Dex)-induced programmed necrosis [32]. Additionally, overexpression of LncRNA NEAT1 regulated steroid-induced necrosis by adsorbing miR-23b-3p [33]. Little is presently noted concerning the performance of LncRNA NR_003508 in tuberculosis. Moreover, the expression of LncRNA NR_003508 was significantly up-regulated with BCG and H37Rv infection; therefore, we further interfered LncRNA NR_003508 to explore the regulatory role in RAW264.7 cells infected with BCG and H37Rv. Our study confirmed that si-NR_003508 decreased the programmed death of cells which showed the same pattern in mice. Of note, the accumulation of mitochondrial ROS also executed programmed necrosis. Once the necrosis pathway is activated, RIPK1-RIPK3 necrosomes stimulate ROS production in mitochondria [34]. Our results also manifested that si-NR_003508 inhibited ROS accumulation, which further validated the above results.

The crosstalk between LncRNAs and miRNAs is common in infectious diseases [35]. Accumulating evidence implied that the mechanisms of LncRNAs were in close contact with the location of LncRNAs in cells [36]. For instance, LncRNAs situated in the cytoplasm may function as miRNA sponges, blocking the effects of miRNAs [37,38]. Usually, LncRNAs relieved the inhibition of downstream target genes by sponging miRNA [39,40,41]. Shen et al. explained in their work that LncRNA Xist could regulate EMT by adsorbing miR-429, thus promoting the invasion and migration of pancreatic cancer [42]. In addition, in mycobacterium tuberculosis-induced infection, LncRNA MIAT regulated autophagy and apoptosis of macrophages through the miR-665/ULK1 signaling axis [43]. Based on the results of FISH, we found LncRNA NR_003508 was mainly located in the cytoplasm, which indicated that LncRNA NR_003508 might also function as a ceRNA. Next, we used bioinformatics analysis to find out the potential binding site between LncRNA NR_003508 and miR-346-3p. Further dual-luciferase reporter assays and FISH assays authenticated that LncRNA NR_003508 could bind to miR-346-3p. As short RNAs with a length of about 22 nucleotides, miRNAs can bind to the target sequence in the 3′-UTR region of mRNA and inhibit mRNA expression [44,45,46]. MiR-346-3p has been reported to be associated with the mTOR signaling pathway and Ca^2+^ overload, which may be related to necrosis [43,47].

In this regard, we further confirmed that LncRNA NR_003508 absorbing miR-346-3p promoted the expression of RIPK1 by binding to the 3′-UTR region. Moreover, a striking elevation of RIPK1 expression was observed in the peripheral blood of TB patients. Additionally, an important feature of programmed necrosis is that it can affect cell membrane permeability. We found that miR-346-3p mimic could inhibit the accumulation of Ca^2+^ concentration in BCG-infected macrophages from inhibiting programmed necrosis. More and more studies confirmed that RIPK1, RIPK3, and MLKL kinases played a central role in TNF-induced programmed necrosis [48,49]. Further experiments indicated that LncRNA NR_003508 sponged miR-346-3p and regulated programmed necrosis through RIPK1/RIPK3/MLKL in RAW264.7 cells. Collectively, these results manifested that LncRNA NR_003508 integrating miR-346-3p activated the expression of RIPK1, RIPK3, and MLKL, then the necrosome facilitated the ROS production of mitochondria and Ca^2+^ influx of cytomembrane in RAW264.7 cells.

Mtb can induce cell necrosis, and cell necrosis is particularly important for the survival of Mtb in the host cells [50]. In the study, we found that LncRNA NR_003508 functioned as a miR-346-3p sponge to positively regulate RIPK1 expression, which promotes programmed necrosis and infection persistence. In fact, therapeutic targeting of long non-coding RNAs (LncRNAs) represents an attractive approach for the treatment of cancer, as well as many other diseases [51]. For example, LncRNA associated with the Progression and Intervention of Atherosclerosis (RAPIA) was highly expressed in advanced atherosclerotic lesions and in macrophages. Additionally, inhibition of the pivotal lncRNA RAPIA may be a novel preventive and therapeutic strategy for advanced atherosclerosis, especially in patients resistant or intolerant to statins [52]. Moreover, LncRNA H19 has a unique expression profile and can act as a sponger of specific miRNAs to regulate the pathogenic process of pancreatic ductal adenocarcinoma and several other types of cancer. H19 may be a novel therapeutic target for pancreatic ductal adenocarcinoma [53]. Based on the above studies, LncRNA NR_003508 small interfering agent, may be used in the clinical treatment of tuberculosis to inhibit the expression of RIPK1 through miR-346-3p, thereby inhibiting programmed necrosis in a certain range and controlling the spread of Mtb in vivo, which might be a novel therapeutic target for tuberculosis.

## 4. Materials and Methods

### 4.1. Cell Culture and Bacterial Strain

RAW264.7 macrophage cell line and 293T cell line were purchased from the Shanghai Institute of Biochemistry and Cell Biology (Shanghai, China), which were cultured in Dulbecco’s modified Eagle medium (DMEM) supplemented with 10% FBS in a humidified condition (37 °C, 5% CO_2_). The BCG and H37Rv were purchased from the Chinese Center for Disease Control and Prevention (Beijing, China), and BCG was grown at 37 °C on Middlebrook 7H9 medium, and H37Rv was grown on acid solid Roche medium. RAW264.7 cells were transfected with small interfering RNA for 24 h and then collected cells after culturing for 18 h at 10 CFU infection by BCG and H37Rv.

### 4.2. Clinical Sample Collection

All procedures were accomplished in accordance with the Declaration of Helsinki. The research was approved by the the Tuberculosis Reference Laboratory, Ningxia Institute for Tuberculosis Control, and the Fourth People’s Hospital of Ningxia Hui Autonomous Region (Yinchuan, Ningxia, China). All patients provided written informed consent. Peripheral blood samples of TB patients (*n*  = 31) and healthy people (*n*  =  31) were collected from Tuberculosis Reference Laboratory, Ningxia Institute for Tuberculosis Control, and the Fourth People’s Hospital of Ningxia Hui Autonomous Region from January 2020 to December 2020, respectively. The age and sex of the participants were random. The expression levels of RIPK1 in blood samples were examined by reverse transcription-polymerase chain reaction (qRT-PCR).

Inclusion criteria of TB patients: ① Tuberculosis was detected in the sputum of the patients, or mycobacterium tuberculosis was positive in vitro culture. ② The sputum smear of the patient is negative, but the imaging, clinical symptoms, and diagnosis of the patient show pulmonary nodular disease.

Exclusion criteria of healthy probands: ① The patient has immune deficiency diseases or other immune system diseases; ② Patients with lung cancer, obstructive pulmonary disease, or other lung diseases; ③ Patients infected with the hepatitis B virus or other bacteria or viruses; ④ Combined with other parts of the junction, cardiac disease or tissue failure.

### 4.3. Ethics and Animals

Twenty specific pathogen-free (SPF) C57BL/6J female mice (6–8 weeks old; 20–25 g) were purchased from Charles River Labs (Beijing, China) and randomly divided into 4 groups. All mice were maintained at 22 ± 2 °C and relative humidity (40–60%), with a 12 h light/dark cycle. The mice had free access to normal food and water throughout the experiment. Fourteen days after intratracheal instillation of BCG (3 × 10^6^ CFU/mouse) or 1×PBS, mice were euthanized in a CO_2_ chamber. All animal experiments were performed according to the guidelines of the Chinese Council on Animal Care and approved by the Ethics Committee for the Conduct of Animal Research of Ningxia University (NXU-ACAU-202009, 27 September 2020).

### 4.4. Transfection of Small Interfering RNA (siRNA) and Plasmid

The si-NR_003508 and negative controls (NC) (NC is a random sequence) were designed and manufactured by GenePharma (Shanghai, China). The plasmid was manufactured by Miaoling (Wuhan, China). RAW264.7 cells were treated with NC and three independent siRNAs, the sequences of which are shown in Table 1. The effective target sequences of siRNA were used. RAW264.7 cells and 293T cells were transfected with siRNAs or plasmid using ZETA LIFE Advanced reagent (San Francisco, CA, USA) at 37 °C. Thirty-six later, the transfected cells were digested and cultivated in cell culture plates for further study.

### 4.5. Everse Transcription-Polymerase Chain Reaction (qRT-PCR)

Total RNA was extracted from RAW264.7 cells using the TRIzol reagent (TIANGEN, Beijing, China). PrimeScript reverse transcriptase (RT) reagent kit (TaKaRa, Shiga, Japan) was used to synthesize LncRNA NR_003508, RIPK1, RIPK3, and MLKL complementary DNA (cDNA) from total RNA. While miRNA was used RevertAid RT Reverse Transcription Kit (Thermo Fisher Scientific, Waltham, MA, USA). Then the program was carried out according to the producer’s recommendations (Abclonal, Wuhan, China). The qRT-PCR reaction mix used the following thermocycling conditions: 95 °C for 30 s (stage 1 for 1 cycle), followed by 40 cycles of 95 °C for 5 s and 60 °C for 30 s, and finally, the dissociation stage. The relative expression levels of LncRNA NR_003508, RIPK1, RIPK3, and MLKL were calculated using the 2^−ΔΔCq^ method and normalized using GAPDH as the reference, while miRNA was normalized using snoRNA202 as the reference [50]. The sequence of the primers is given in Table 2.

### 4.6. Western Blot

The RAW264.7 cells were seeded on a 6-well plate at 1 × 10^6^ cells/well, washed three times with 1×PBS, and then lysed in protein extraction buffer (KeyGEN BioTECH, Nanjing, China) for 20 min to extract total protein. The protein concentration was determined by BCA Assay Kit (Thermo Fisher Scientific, Waltham, MA, USA). Subsequently, the equal 30 μg proteins from each sample were probed to SDS-PAGE and transferred to PVDF membrane (wet transfer method), and the percentage of the resolving or separating gel was 10%. The membranes were blocked with 5% nonfat milk at room temperature for 2 h and then incubated with the primary antibodies overnight at 4 °C. After incubation with the secondary antibody at room temperature for 2 h, the protein bands were visualized by ECL (Thermo Fisher Scientific, Waltham, MA, USA) and the results were analyzed by Image J. The GAPDH (Proteintech; 60004-1-Ig), RIPK1 (Proteintech; 17519-1-AP), RIPK3 (Cell Signaling Technology; 15828S), p-MLKL (Affinity; AF7420), MLKL (Abclonal; WH196063), p-RIPK1 (Abmart; TA2398S), p-RIPK3 (Abmart; TA3900S), the goat anti-rabbit IgG secondary antibody and the goat anti-mouse IgG secondary antibody were used for the Western blot analyses.

### 4.7. Immunofluorescence Assay

RAW264.7 cells were seeded on a 12-well plate at 5 × 10^5^ cells/well and treated with or without BCG for 18 h, incubated overnight. The cells were fixed with 4% paraformaldehyde for 30 min at room temperature and permeabilized for 30 min with a treatment in PBS containing 0.5% Triton x-100, followed by blocking with 3% BSA (Thermo Fisher Scientific, Waltham, MA, USA) for 1 h. The cells were then incubated with primary antibody for 2 h at 37 °C followed by the appropriate secondary antibody (Abcam; ab150077). The nucleus was stained with DAPI (ZSGB-BIO, Beijing, China) for 10 min. Images were obtained with a laser confocal microscope (Olympus, Tokyo, Japan).

### 4.8. PI Staining and Confocal Microscope Assay for Cell Necrosis

In order to study necrosis, RAW264.7 cells were plated in 12-well plates at 5 × 10^5^ cells/well. After being infected with BCG for 18 h, the cells were gently washed using 1×PBS solution and fixed in 4% formaldehyde for 20 min. For extracellular staining, the cells were incubated for 30 min in 10 μg/Ml PI working solution (1 Μl/well). Excess staining was rinsed with 1×PBS, and images were acquired using a fluorescence microscope (Olympus, Tokyo, Japan).

### 4.9. Fluorescence in Situ Hybridization (FISH)

The FISH assay was performed to identify the subcellular location of LncRNA NR_003508, miR-346-3p, and RIPK1. The experiment was conducted according to the instructions of the fluorescent in situ hybridization kit (GenePharma, Shanghai, China). Briefly, RAW264.7 cells were grown on the slides in 12-well plates at 5 × 10^5^ cells/well, washed with PBS, and fixed in 4% paraformaldehyde. Then, the cells were permeabilized with 0.1% TritonX-100 at room temperature for 15 min, followed by incubation with hybridization buffer at 37 °C overnight. Afterward, the nucleus was stained with DAPI for 10 min after washing by a hybridization buffer. The cells were visualized by a fluorescence microscope (Olympus, Tokyo, Japan).

### 4.10. Annexin V and PI Staining for Apoptosis and Necrosis by Flow Cytometry

RAW264.7 cells were stained with Annexin V-FITC and propidium iodide (PI) (BD Biosciences Clontech, NJ, USA). After being treated with BCG and si-NR_003508 or miRNA mimics, the samples were washed twice and adjusted to the appropriate concentration with 1×PBS. Then, 500 μL of suspension was added to each labeled tube, and 2 μL annexin V-FITC and 2 μL PI were added into the labeled tube and cultivated for 30 min at 37 °C in the dark. The necrosis rate was analyzed by FCM.

### 4.11. Determination of the ROS and Ca^2+^ Level

RAW264.7 cells were treated with BCG and si-NR_003508 or miRNA mimics. The OxiSelectIn Vitro ROS/RNS Assay Kit (Cell Biolabs, San Diego, CA, USA) was used to detect the level of ROS in cell samples. A cell calcium detection kit (FluO-3 Probe Method) (Bisantrene, Shanghai, China) was adopted to measure the Ca^2+^ levels in RAW264.7 cells. Additionally, the ROS and Ca^2+^ levels in each group were detected by flow cytometry and analyzed in triplicate using commercial kits according to the manufacturer’s instructions.

### 4.12. Luciferase Reporter Assay

To construct the reporter vectors, the wild type (WT) and mutant (Mut) of the LncRNA NR_003508 and RIPK1 sequence containing the binding site of miR-346-3p were cloned into PPGk luciferase reporter vector (Miaoling Biotechnology, Wuhan, China). miR-346-3p mimic was manufactured by GenePharma (Shanghai, China), and the reporter vectors were co-transfected into RAW264.7 cells using ZETA LIFE Advanced reagent (San Francisco, CA, USA) for 36 h. Dual-Luciferase Reporter Assay System (Promega, Madison, WI, USA) was used to detect and analyze the relative luciferase activity.

### 4.13. Hematoxylin-Eosin (H&E) Staining 

The mouse lung tissues were fixed in 4% formaldehyde at room temperature for 48 h. After washing in fresh PBS, fixed tissues were dehydrated, cleared, and embedded in paraffin by routine methods. All lung tissues were sectioned into thin pieces (7 μm in thickness), deparaffinized, and stained with H&E reagents at room temperature for 3 min and then visualized under an optical microscope (Tokyo, Japan).

### 4.14. Transmission Electron Microscopy (TEM) Analysis

The ultrastructure of RAW264.7 cells was examined through TEM. The cells were digested with pancreatin and treated with 3% glutaraldehyde at room temperature for at least 2 h. After the glutaraldehyde solution was discarded, the samples were washed three times with 1×PBS for 15 min and fixed with 1% osmium acid for 1 h. After gradient dehydration with ethanol, the cells were embedded in epoxy resin and cut into 70 nm sections. Ultrathin sections were stained with uranyl acetate and lead citrate and examined with transmission electron microscopy (Tokyo, Japan).

### 4.15. Statistical Analysis

Statistical analysis was performed by GraphPad Prism 9 software (Graph Pad, USA). All data were obtained from at least three independent experiments and expressed as mean ± SEM. Two groups were compared by *t*-test. The data from experiments with three or more groups were compared using the one-way ANOVA. For all tests, a *p*-value < 0.05 was assigned to be statistically significant.

## Figures and Tables

**Figure 1 ijms-24-08016-f001:**
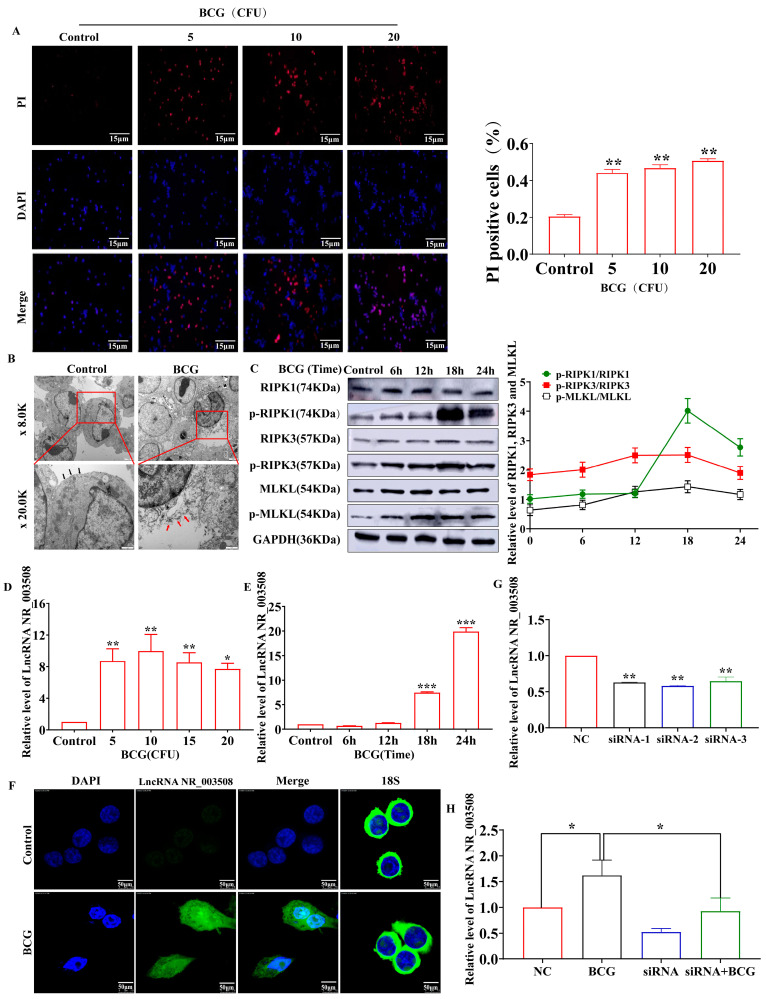
BCG-induced RAW264.7 programmed necrosis and up−regulated LncRNA NR_003508 expression: (**A**) The necrosis rate of cells was measured using PI staining. Left, the histogram. (scale bar =15 μm). ** *p* < 0.01 vs. the control cells. (**B**) Ultrastructural changes in BCG−treated RAW264.7 cells were observed by electron microscopy (*n* = 3, magnification, 8.0K, scale bar = 2 μm and 20.0K, scale bar = 1 μm). Black arrows denote normal cell membranes. Red arrows indicate the release of cellular contents when the cell membrane integrity is ruptured. (**C**) The RIPK1, p−RIPK1, RIPK3, p−RIPK3, MLKL and p−MLKL expression levels infected with BCG at varied time points were detected by Western blot (*n* = 3). The broken line graph was calculated by the gray value of p−RIPK1/RIPK1, p−RIPK3/RIPK3 and p−MLKL/MLKL (*n* = 3). * *p* < 0.05, ** *p* < 0.01, *** *p* < 0.001 vs. the control cells. (**D**) qRT−PCR detected the LncRNA NR_003508 expression infected with BCG for different CFU (*n* = 3). * *p* < 0.05, ** *p* < 0.01 vs. the control cells. (**E**) The detection of cellular LncRNA NR_003508 expression followed by BCG at 10 CFU for varied time points by qRT−PCR assay (*n* = 3). *** *p* < 0.001 vs. the control cells. (**F**) FISH was used to detect the expression of LncRNA NR_003508 in RAW264.7 cells with BCG-infected for 18 h. (*n* = 3, scale bar = 50 μm). (**G**) qRT−PCR was performed to measure the interference efficiency of si−NR_003508 (*n* = 3). ** *p* < 0.01 vs. the NC group. (**H**) LncRNA NR_003508 expression of the RAW264.7 cells treated with BCG/siRNA was detected by qRT−PCR (*n* = 3). * *p* < 0.05 vs. the NC group, * *p* < 0.05 vs. the BCG group. The data were presented as mean ± SEM, and the representativeness of three independent experiments was shown (*n* = 3, each group).

**Figure 2 ijms-24-08016-f002:**
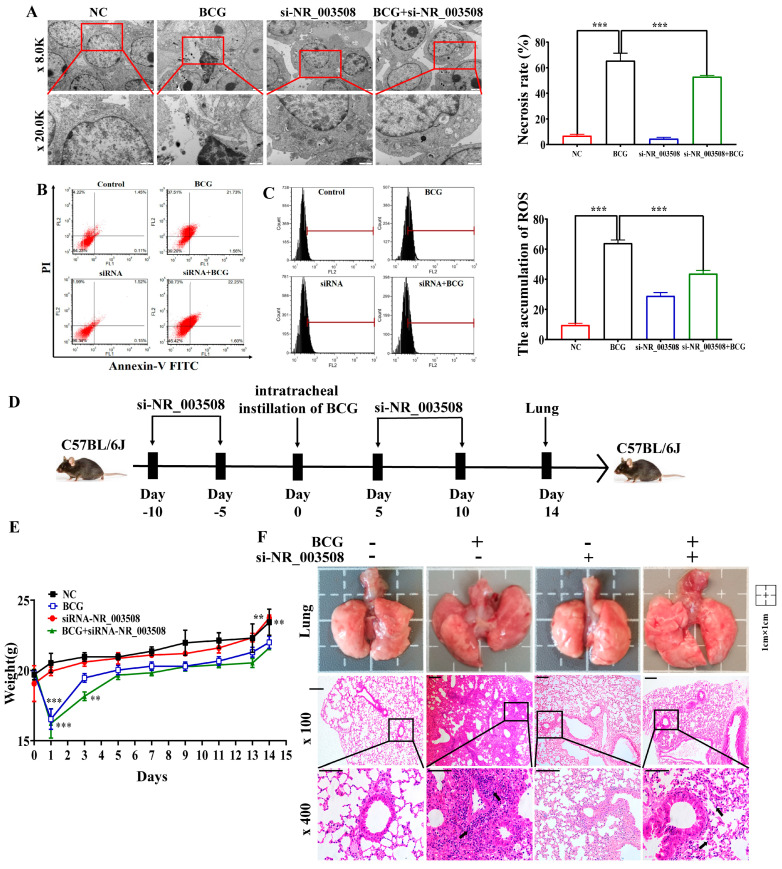
LncRNA NR_003508 alleviated BCG-induced cell-programmed necrosis and lung injury in mice: (**A**) The cell changes and necrosis morphology of RAW264.7 cells were measured by transmission electron microscopy (TEM) under 8.0K (scale bar = 2 μm) and 20.0K (scale bar = 1 μm) magnification (*n* = 3). (**B**) The flow cytometry was used to detect the necrosis rate using Annexin V−FITC/PI staining in RAW264.7 (*n* = 3). ** *p* < 0.01 vs. the NC group. (**C**) CellROX Orange Reagent detected ROS accumulation in RAW264.7 (*n* = 3). ** *p* < 0.01 vs. the NC group. (**D**) The schematic diagram of mice experiments. (**E**) The mice’s body weight in each group (*n* = 3). ** *p* < 0.01, *** *p* < 0.001 vs. the control cells. (**F**) Epigenetic changes in lung and H&E staining were performed on lung tissue slices 14 d after BCG infection. Black arrows indicate increased infiltration of inflammatory cells. (magnification, ×100, scale bar = 200 μm and ×400, scale bar = 100 μm). The data were based on at least three independent experiments presented as mean ± SEM (*n* = 3, each group).

**Figure 3 ijms-24-08016-f003:**
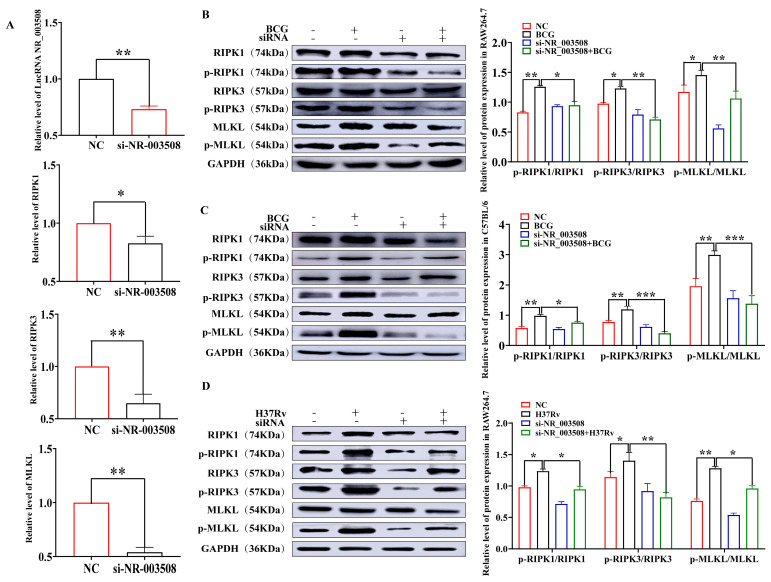
LncRNA NR_003508 promoted necrosis-associated-proteins expression in RAW264.7 cells and C57BL/6J mice infected with BCG: (**A**) The expression of Lnc NR_003508, RIPK1, RIPK3, and MLKL with si−NR_003508 transfection was detected by qRT−PCR assay (*n* = 3). * *p* < 0.05, ** *p* < 0.01 vs. the NC group. The expression of p−RIPK1/RIPK1, p−RIPK3/RIPK3, and p−MLKL/MLKL in RAW264.7 cells (**B**) and mouse lung tissues (**C**) with si−NR_003508 transfection and BCG infection was detected by Western blot (*n* = 3). * *p* < 0.05, ** *p* < 0.01 vs. the NC group, * *p* < 0.05, ** *p* < 0.01, *** *p* < 0.001 vs. the BCG group. (**D**) Western blot was used to detect the expression of p−RIPK1/RIPK1, p−RIPK3/RIPK3, and p−MLKL/MLKL in RAW264.7 with si−NR_003508 transfection and H37Rv infection (*n* = 3). * *p* < 0.05, ** *p* < 0.01 vs. the NC group, * *p* < 0.05, ** *p* < 0.01 vs. the H37Rv group. Three independent experiments were performed (*n* = 3, each group).

**Figure 4 ijms-24-08016-f004:**
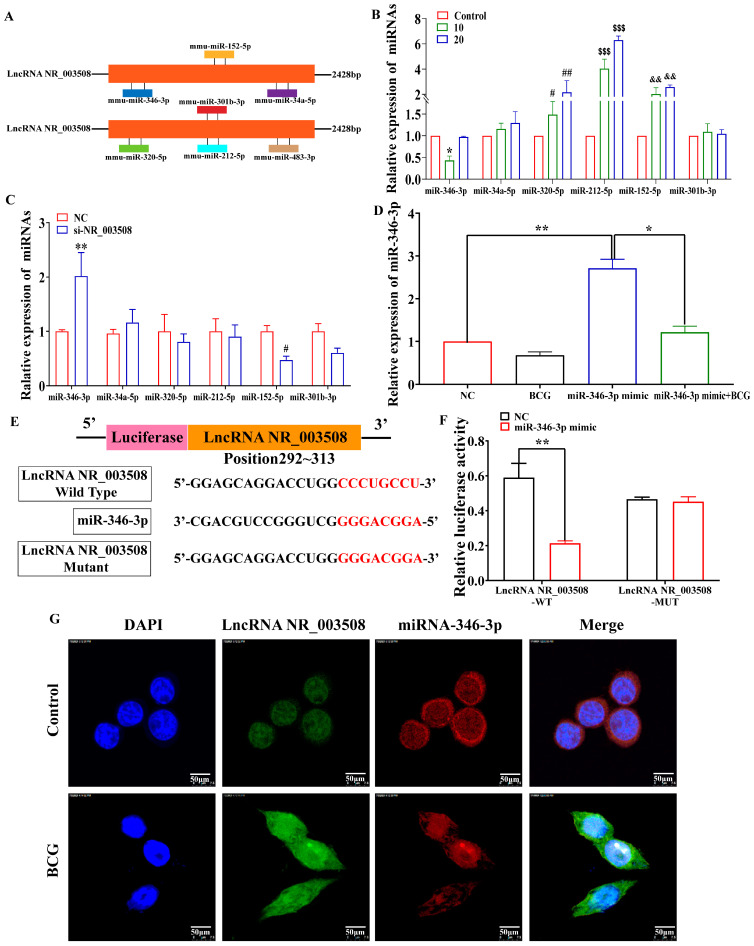
LncRNA NR_003508 functioned as a ceRNA and regulated miR−346−3p expression: (**A**) The BiBiServ2-RNAhybrid accessed on 10 July 2019 and TargetScan 8.0 accessed on 29 September 2021 softwares predicted the interactions between LncRNA NR_003508 and miRNAs associated with necrosis. (**B**) The relative expression of miRNAs was detected using qRT−PCR infected at 10 or 20 CFU (*n* = 3). *^,#^
*p* < 0.05, ^##,&&^
*p* < 0.01, ^$$$^
*p* < 0.001 vs. the control cells. qRT−PCR detected the expression of miRNAs in si−NR_003508 and NC groups (**C**) (*n* = 3), ^#^
*p* < 0.05, ** *p* < 0.01 vs. the NC group as well as the miR-346-3p expression in RAW264.7 (**D**) (*n* = 3). * *p* < 0.05 vs. the miR−46−3p mimic group, ** *p* < 0.01 vs. the NC group. (**E**) The binding sites between LncRNA NR_003508 and miR−346−3p were predicted by Bielefeld Bioinformatics Service accessed on 10 July 2019. (**F**) The interaction between LncRNA NR_003508 and miR−346−3p was verified by using luciferase reporter (*n* = 3). ** *p* < 0.01 vs. the NC group. (**G**) Immunofluorescence measured the expression of LncRNA NR_003508 and miR−346−3p with transfecting BCG and miR−346−3p mimic in RAW264.7 cells (*n* = 3). (Scale bar = 50 μm). Three independent experiments were performed (*n* = 3, each group).

**Figure 5 ijms-24-08016-f005:**
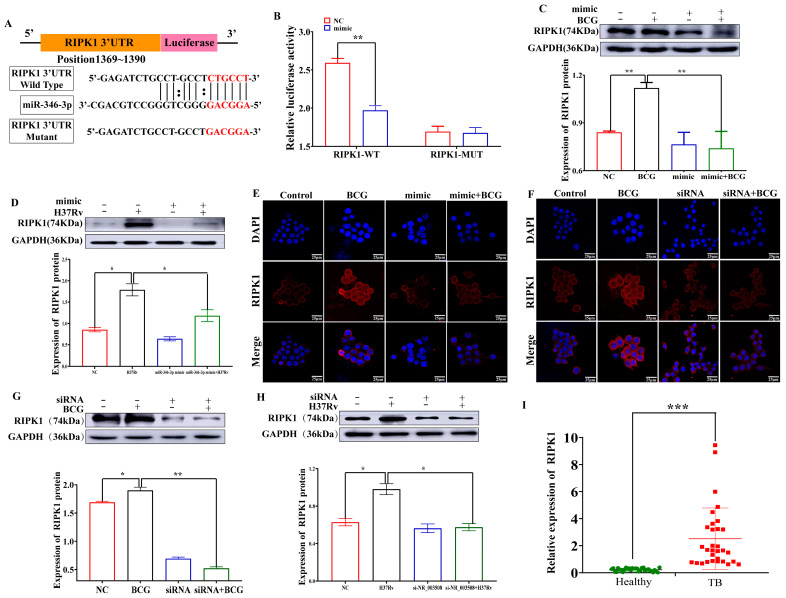
LncRNA NR_003508 regulated RIPK1 expression by binding to miR−346−3p: (**A**) The online prediction software TargetScan 8.0, accessed on 29 September 2021, predicted binding sites between miR−346−3p and RIPK1. (**B**) Luciferase reporter assay was employed to monitor the relative luciferase activity (*n* = 3). ** *p* < 0.01 vs. the NC group. Western blot was used to detect the expression of RIPK1 with BCG (**C**) (*n* = 3), ** *p* < 0.01 vs. the NC group, ** *p* < 0.01 vs. the BCG group, and H37Rv infection (**D**) (*n* = 3), * *p* < 0.05 vs. the NC and H37Rv group. Immunofluorescence was utilized for RIPK1 expression with miR−346−3p mimic (**E**) and si−NR_003508 transfection (**F**) (*n* = 3) (Scale bar = 25 μm). The level of RIPK1 with BCG (**G**) (*n* = 3), * *p* < 0.05 vs. the NC group, ** *p* < 0.01 vs. the BCG group, and H37Rv infection (**H**) (*n* = 3), * *p* < 0.05 vs. the NC and H37Rv group was measured by Western blot. (**I**) The RIPK1 expression in peripheral blood of TB patients was detected by qRT−PCR (*n* = 3), *** *p* < 0.001 vs. the Healthy group. Data were based on at least three independent experiments (*n* = 3, each group).

**Figure 6 ijms-24-08016-f006:**
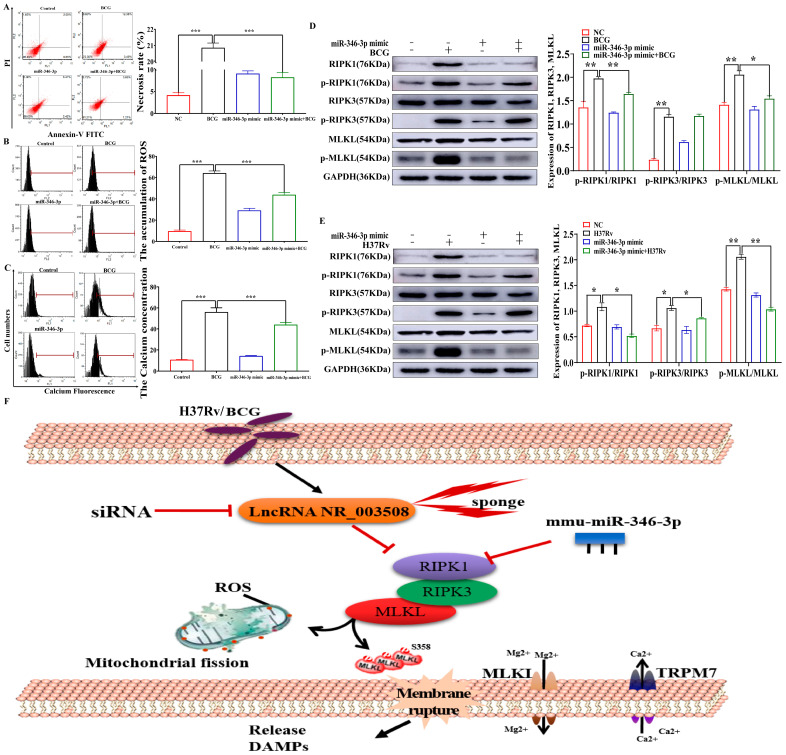
MiR−346−3p participated in programmed necrosis through RIPK1/RIPK3/MLKL axis: (**A**) The necrosis rate was determined by AnnexinV−FITC (AV) and PI staining. *** *p* < 0.001 vs. the NC group, *** *p* < 0.001 vs. the BCG group. (**B**) The ROS accumulation was determined by flow cytometry in RAW264.7 cells, and quantitative analysis was performed by FACS assay (*n* = 3). *** *p* < 0.001 vs. the NC group, *** *p* < 0.001 vs. the BCG group. (**C**) Flow cytometry was used to detect the intracellular Ca^2+^ levels in macrophages. Right, the quantitative analysis results (*n* = 3). *** *p* < 0.001 vs. the NC group, *** *p* < 0.001 vs. the BCG group. The protein expression levels of RIPK1, p−RIPK1, RIPK3, p−RIPK3, MLKL, and p−MLKL with BCG (**D**) ** *p* < 0.01 vs. the NC group, * *p* < 0.05, ** *p* < 0.01 vs. the BCG group and H37Rv (**E**) * *p* < 0.05, ** *p* < 0.01 vs. the NC group, * *p* < 0.05, ** *p* < 0.01 vs. the H37Rv group infection were detected by Western blot. The histogram was manufactured by the gray values of p−RIPK1/RIPK1, p−RIPK3/RIPK3, and p−MLKL/MLKL (*n* = 3). (**F**) Schematic diagram of LncRNA NR_003508 regulated RIPK1−mediated programmed necrosis pathway via miR−346−3p. Three independent experiments were performed (*n* = 3, each group).

**Table 1 ijms-24-08016-t001:** The small interferences sequence.

	Sense (5′-3′)	Antisense (5′-3′)
siRNA-1	GCGUUGAUUCAGUCAACUUTT	AAGUUGACUGAAUCAACGCTT
siRNA-2	GCACACGGUCACUGAAAUUTT	AAUUUCAGUGACCGUGUGCTT
siRNA-3	ACACUGCUAGAAAUAAAUUTT	AAUUUAUUUCUAGCAGUGUGC

**Table 2 ijms-24-08016-t002:** The primers sequence.

Primer	Sequence (5′-3′)
LncRNA NR_003508	F: GTATGAGGAGAAGGTGCGGC
	R: CCAGAACTCTGGTCCCCAAT
RIPK1	F: AGAACAACCTGGATCGCTGC
	R: CCTGCACACTGCGATCATT
RIPK3	F: CCTTCAGAGGCACAACACCT
	R: TGTCATTGGATTCGGTGGG
MLKL	F GCCACTGGAAAGATCCCATTTG
	R: TTCCCGCAACAACTCAGG
GAPDH	F: TACCCCCAATGTGTCCGTC
	R: AAGAGTGGGAGTTGCTGTTGAAG
miR-346-3p	RT: CTCAACTGGTGTCGTGGAGTCGGCAATTCAGTTGAGGCTGCAGGC
	F: ACACTCCAGCTGGCAGGCAGGGGCTGGGCCTG
miR-34a-5p	RT: CTCAACTGGTGTCGTGGAGTCGGCAATTCAGTTGAGGCTGCAGGC
	F: ACACTCCAGCTGGCCGGGAACGTCGAGACTGG
miR-320-5p	RT: CTCAACTGGTGTCGTGGAGTCGGCAATTCAGTTGAGCGGAAGAGA
	F: ACACTCCAGCTGGCTTCCCTTTGTCATCCTTT
miR-212-5p	RT: CTCAACTGGTGTCGTGGAGTCGGCAATTCAGTTGAGTGGAACCGA
	F: ACACTCCAGCTGGCAGGGGCTGGCTTTCCTCT
miR-152-5p	RT: CTCAACTGGTGTCGTGGAGTCGGCAATTCAGTTGAGTCCAAGACA
	F: ACACTCCAGCTGGCAGAGGGTGCGCCGGGCCCT
miR-301b-3p	RT: CTCAACTGGTGTCGTGGAGTCGGCAATTCAGTTGAGGTCACGTTA
	F: ACACTCCAGCTGGCAGGAAGCCCTGGAGGGGCT
miRNA Universal Reverse sequence	TGGTGTCGTGGAGTCG
SnoRNA202	RT: CTCAACTGGTGTCGTGGAGTCGGCAATTCAGTTGAGCCTGGAAT
	F: ACACTCCAGCTGGCAGTCAGGCTCCTGGCTAGA

## Data Availability

All data used and analyzed during this study are available from the corresponding authors with a reasonable request.

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
