# Peer review of "LncRNA NR_003508 Suppresses Mycobacterium tuberculosis-Induced Programmed Necrosis via Sponging miR-346-3p to Regulate RIPK1"

_ijms, 2023, doi:10.3390/ijms24098016_

Round 1
Reviewer 1 Report
The manuscript entitled “LncRNA NR_003508 suppresses Bacillus Calmette-Guerin-induced programmed necrosis via sponging miR-346-3p to regu- 3 late RIPK1” by Liu et al is a good piece of work. The experiments are exhaustive and propagate the field forward.
Specific comments below:
- Please enhance the quality of the images. It is hard to understand the images and interpret them at times:
- Figure 2B Dual staining of Annexin A2 and PI. I do see any necrosis in the histogram. They all look alike. Could you enhance the file? Most of the cells do not show necrosis.
- Please highlight the rationale for using the RAW 264.7 cell line. It is a mouse macrophage cell line. If you want to study programmed necrosis, the cancer cell lines would be more appropriate.
- Please discuss your findings in more detail. Your line “This study elucidated that lncRNA NR_003508 might 338 be a novel therapeutic target for tuberculosis” appears to be important. Could you elaborate on it a bit more?
Reviewer 2 Report
Various cell death modalities may result from either mycobacteria-induced events or host-induced innate defense mechanisms. The dynamics of these modalities determine the outcome of Mtb infection. Several host factors are involved in multiple cell death pathways, and Mtb factors can inhibit host defense mechanisms at various levels. The study under review is focused on RNA-mediated mechanisms of macrophage death modalities regulation in TB. Though the study is very interesting in itself, there are some major concerns:
1. Materials and Methods section should be supplemented with the description of TB patients (inclusion/exclusion criteria) and healthy probands (exclusion criteria).
2. Usage of GAPDH gene as a reference for RT-PCR experiments is questionable. Multiple studies have demonstrated that the best choice of reference gene in the analysis of the relative miRNA expression level is either a miRNA or the geometric mean of several stably expressed miRNAs. Additionally, it is recommended that a reference gene be searched for in each experimental system (Vandesompele J, De Preter K, Pattyn F, Poppe B, Van Roy N, De Paepe A, et al. Accurate normalization of real-time quantitative RT-PCR data by geometric averaging of multiple internal control genes. Genome Biol. 2002; 3(7): research0034. doi: 10.1186/gb-2002-3-7-research0034) ( Wong L, Lee K, Russell I, Chen C (2007) Endogenous Controls for Real-Time Quantitation of miRNA Using TaqMan® MicroRNA Assays. Applied Biosystems Application Note, Publication 127AP11-01, available at www.appliedbiosystems.com).
3. Table 2 (Methods) is a mess.
4. The cell death modalities induced by virulent Mycobacterium tuberulosis and avirulent M. bovis BCG differ. Thus the authors should be more causious in transferring results obtained with the cell line and BCG in vitro to human TB.
Round 2
Reviewer 1 Report
Dear authors,
I did not receive the diagrams in the file that I received.
Please contact the editor to provide the figures in the revised version.
I have written to the editor to provide me the diagrams, to help me making my final decision.
Reviewer 2 Report
All my concerns answered, the article might be published. Still minor spell check is required.
Round 3
Reviewer 1 Report
Line 455: Dear Authors,
You double-stained the cells with Annexin V and Propidium iodide, hence write the subheading as
“Annexin V and PI Staining for Apoptosis by Flow Cytometry”. Not just PI Staining.
Also, cite
Selective phenylalanine to proline substitution for improved antimicrobial and anticancer activities of peptides designed on phenylalanine heptad repeat. Acta Biomater. 2017 Jul 15;57:170-186. doi: 10.1016/j.actbio.2017.05.007. Epub 2017 May 5. PMID: 28483698. At the end of the paragraph.
Many congratulations on improving the work!
